# AdvWeb: Controllable Black-box Attacks on VLM-powered Web Agents

## Abstract

Vision Language Models (VLMs) have revolutionized the creation of generalist web agents, empowering them to autonomously complete diverse tasks on real-world websites, thereby boosting human efficiency and productivity. However, despite their remarkable capabilities, the safety and security of these agents against malicious attacks remain critically underexplored, raising significant concerns about their safe deployment. To uncover and exploit such vulnerabilities in web agents, we provide AdvWeb, a novel black-box attack framework designed against web agents. AdvWeb trains an adversarial prompter model that generates and injects adversarial prompts into web pages, misleading web agents into executing targeted adversarial actions such as inappropriate stock purchases or erroneous bank transactions—actions that could lead to severe consequences. With only black-box access to the web agent, we train and optimize the adversarial prompter model using Direct Policy Optimization (DPO), leveraging both successful and failed attack strings against the target agent. Unlike prior approaches, our adversarial string injection maintains stealth and control: (1) the appearance of the website remains unchanged before and after the attack, making it nearly impossible for users to detect tampering, and (2) attackers can modify specific substrings within the generated adversarial string to seamlessly change the attack objective (e.g., purchasing stocks from a different company), greatly enhancing attack flexibility and efficiency. We conduct extensive evaluations, demonstrating that AdvWeb achieves high success rates in attacking state-of-the-art GPT-4V-based VLM agents across various web tasks in black-box settings. Our findings expose critical vulnerabilities in current LLM/VLM-based agents, emphasizing the urgent need for developing more reliable web agents and implementing effective defenses against such adversarial threats.

## 1 Introduction

The rapid evolution of Large Language Models (LLMs) and Vision Language Models (VLMs) has enabled the development of generalist web agents, which are capable of autonomously interacting with real-world websites and executing tasks, such as making stock purchases, performing healthcare operations, or handling financial transactions (Nakano et al., 2021; Wu et al., 2024b; Yao et al., 2022). These agents, by leveraging tools, APIs, and complex website interactions, hold tremendous potential for enhancing human productivity across various domains like finance, healthcare, and e-commerce (Zhou et al., 2023; Deng et al., 2024; Zheng et al., 2024). Despite their success, these agents also bring unprecedented security challenges, especially in terms of their robustness to malicious adversarial attacks, which remains underexplored in existing literature.

Recent works have proposed adversarial attacks against generalist web agents to uncover vulnerabilities before real-world deployment (Yang et al., 2024; Wang et al., 2024; Wu et al., 2024a). However, existing approaches are either limited by high attack costs, requiring human effort in manually designing the attack prompts (Wu et al., 2024b; Liao et al., 2024), or focused primarily on individual attack scenarios (Mo et al., 2024), leaving gaps in developing more efficient and adaptable attack frameworks for web agents. Many adversarial attacks against LLMs and VLMs have also been proposed to automatically optimize the attack prompts (Guo et al., 2024; Huang et al., 2024). However, they can not be flexibly adapted to attack VLM-based agents (Zou et al., 2023) and struggle to achieve transferability to our black-box attack setting (Liu et al., 2024c).

Figure 1: **Overview of `AdvWeb`.** We train an adversarial prompter model to generate adversarial strings added to the website. The injected string is hidden in invisible HTML fields and does not change the website rendering. Web agents working on the injected malicious website will be mistled to perform targeted actions: buy Microsoft stocks can be attacked to buy NVIDIA stocks instead, leading to severe consequences.

To address these challenges, we propose `AdvWeb`, a novel black-box controllable attacking framework specifically designed to exploit vulnerabilities in generalist web agents. Our approach generates and injects invisible adversarial strings into web pages, misleading the agents into performing targeted adversarial actions, such as executing erroneous financial transactions or purchasing incorrect stocks, which can lead to significant consequences. By using Direct Policy Optimization (DPO) (Rafailov et al., 2024), `AdvWeb` optimizes adversarial string generation based on both successful and unsuccessful attacks against the black-box web agent, allowing for efficient and flexible attacks. Notably, `AdvWeb` enables attackers to easily control and modify adversarial strings without re-optimizing them, making it possible to achieve different attack goals, such as targeting different companies or actions, with minimal additional effort.

To evaluate the effectiveness of `AdvWeb`, we test our approach extensively against SeeAct (Zheng et al., 2024), a state-of-the-art (SOTA) VLM-based web agent framework, on various web tasks in a black-box setting. Our results demonstrate that `AdvWeb` is highly effective, achieving a 97.5% attack success rate against GPT-4V-based SeeAct across different website domains and various tasks, significantly outperforming baseline methods. Our attacks also exhibit strong controllability, with a 98.5% attack success rate after changing the attack targets without further optimizations. Experiments also show that our stealthy attack strings can be flexibly hidden in different HTML fields while maintaining high attack success rates. These findings highlight the vulnerability of current LLM/VLM-based agents and underscore the need for developing more robust defenses to safeguard their deployment in the real world.

Our key contributions are summarized as follows: (1) We propose `AdvWeb`, the first black-box targeted attacking framework against VLM-based web agents, which trains a generative model to automatically generate adversarial prompts injected into the HTML contents. (2) We propose a two-stage training paradigm that incorporates reinforcement learning (RL) from the black-box feedback of the victim agents to optimize the adversarial string. (3) We perform real-world attacks against SOTA web agent on 440 tasks across 4 different domains. We show that our attack is effective, achieving an attack success rate of 97.5%. Our adversarial strings are also highly controllable, with a 98.5% transfer-based attack success rate to different attack targets. (4) We conduct a series of ablation studies and show that the proposed training framework is crucial and effective for black-box attacks. Our generated adversarial strings can also be robustly adapted to different attack settings, achieving near 97.0% attack success rate when we vary different HTML fields.

## 2 RELATED WORK

**Adversarial Attack on LLM.** Many approaches have been proposed to jailbreak aligned LLMs, encouraging them to generate harmful content or answer malicious questions. Due to the discrete nature of tokens, optimizing these attacks is more challenging than in image-based attacks (Carlini et al., 2024). Early works (Ebrahimi et al., 2018; Wallace et al., 2019; Shin et al., 2020) optimize

input-agnostic token sequences to elicit specific predictions or generate harmful outputs, leveraging greedy search or gradient information to modify influential tokens. Later, ARCA (Jones et al., 2023) refines these techniques by simultaneously assessing the impact of multiple token swaps. The GCG Attack (Zou et al., 2023) then successfully optimized suffixes to elicit affirmative responses, making attacks more effective. However, the adversarial strings generated by all previous works are unreadable and are easily detected by perplexity-based detectors. AutoDan (Liu et al., 2024c) improves the stealthy of the generated adversarial prompts by leveraging a carefully designed hierarchical genetic algorithm that maintains semantic meaningfulness. Additionally, AmpleGCG (Liao & Sun, 2024) and AdvPrompter (Paulus et al., 2024) directly employ generated models to generate adversarial suffixes without relying on gradient-based optimization. However, these attacks are primarily enforced towards **simple objectives** (e.g., eliciting confirmative responses to harmful queries) and are no longer effective against more complex attack objectives on VLM-powered web agents. To address this limitation, we present the first attack framework capable of handling **diverse and complex objectives** (e.g., manipulating a stock purchase decision) while maintaining stealthiness and controllability.

**Web Agents.** As LLMs (Brown et al., 2020; Achiam et al., 2023; Touvron et al., 2023) and VLMs (Liu et al., 2024b; Dubey et al., 2024; Team et al., 2023) rapidly evolve, their capabilities have significantly expanded, particularly in leveraging visual perception, complex reasoning, and planning to assist with daily tasks. Some works (Nakano et al., 2021; Wu et al., 2024b) build generalist web agents by leveraging the LLMs augmented with retrieval capabilities over the websites, which is useful for information seeking. Recent works (Yao et al., 2022; Zhou et al., 2023; Deng et al., 2024) have developed web agents that take raw HTML content as input and can directly perform tasks in simulated or realistic web environments based on human instructions. However, HTML content can be noisier compared to the rendered visuals used in human web browsing and provides lower information density, which leads to low task success rates and limited deployment in practice. To fully leverage the model capabilities, SeeAct (Zheng et al., 2024) proposes a generalist web agent framework that consists of a two-stage pipeline and incorporates rendered screenshots as input, yielding stronger reasoning and achieving SOTA task completion performances. Therefore, in this work, we focus on attacking SeeAct as our target agent. However, it is important to note that our proposed attack strategies are readily applicable to all web agents that use webpage screenshots and/or HTML content as input.

**Existing Attacks against Web Agents.** To the best of our knowledge, there exists only a limited body of research examining potential attacks against web agents. Yang et al. (2024) and Wang et al. (2024) investigate the insertion of backdoor triggers into web agents through fine-tuning backbone models with white-box access, aiming to mislead agents into making incorrect purchase decisions. Wu et al. (2024b) and Liao et al. (2024) manipulate the web agents by injecting malicious instructions into the web contents, misleading the agent to execute the indirect prompts, leading to wrong results or privacy leakage. However, the malicious instructions are manually designed and written with heuristics, leading to limited scalability and flexibility. Wu et al. (2024a) shares a similar spirit with us by focusing on automatically optimizing adversarial input to mislead the web agents. However, they either require white-box access to the target agent to perform gradient-based optimization or have limited attack success rates by transferring successful attacks on multiple CLIP models to proprietary VLM-based agents. In contrast, our work attacks the web agents in a black-box setting. By leveraging reinforcement learning to learn from both positive and negative feedback of the black-box model, we train a generative model to generate the adversarial strings that can efficiently and flexibly attack the web agents to perform targeted actions.

## 3 TARGETED BLACK-BOX ATTACK AGAINST WEB AGENTS

### 3.1 PRELIMINARIES ON WEB AGENT FORMULATION

Web agents, like SeeAct (Zheng et al., 2024), are designed to autonomously interact with websites and execute tasks based on user requests. Given a specific website (e.g., a stock trading platform) and a task request $T$ (e.g., "buy one Microsoft stock"), the web agent must generate a sequence of executable actions $\{a_1, a_2, \ldots, a_n\}$ to successfully complete the task $T$ on the target website.

At each time step $t$, the agent derives the action $a_t$ based on the current environment observation $s_t$, the previously executed actions $A_t = \{a_1, a_2, \ldots, a_{t-1}\}$, and the task $T$. For the SeeAct agent, the observation $s_t$ consists of two components: the HTML content $h_t$ of the webpage and the cor-

responding rendered screenshot $i_t = I(h_t)$, and the agent utilizes an VLM (e.g., GPT-4V) as its backend policy model $\Pi$ to generate the corresponding action, as shown in the following equation:

$$a_t = \Pi(s_t, T, A_t) = \Pi(\{i_t, h_t\}, T, A_t) \tag{1}$$

Each action $a_t$ is formulated as a triplet $(o_t, r_t, e_t)$, where $o_t$ specifies the operation to perform, $r_t$ represents a corresponding argument for the operation, and $e_t$ refers to the target HTML element. For example, to fill in the stock name on the trading website, the agent will type ($o_t$) the desired stock name, Microsoft ($r_t$), into the stock input text box ($e_t$). Once the action $a_t$ is performed, the website updates accordingly, and the agent continues this process until the task is completed. For brevity, we omit the time-step notion $t$ in subsequent equations unless otherwise stated.

## 3.2 THREAT MODEL

**Attack Objective.** We consider targeted attacks against the web agents that change the agent's action to a targeted adversarial action $a_{adv} = (o, r_{adv}, e)$ that contains a targeted adversarial argument while keeping the operation the same. This attack can lead to severe consequences since the agent will proceed with the normal operation but with a wrong malicious argument. For example, a request to buy Microsoft ($r$) stocks can be attacked to buy NVIDIA ($r_{adv}$) stocks instead, leading to huge financial losses to the user.

**Environment Access and Attack Scenarios.** Since most web agents are powered by proprietary VLMs, we consider the black-box setting where the attacker does not have access to the agent framework, the backend model weights, or the backend model logits. The attacker only has access to the HTML content $h$ in the website, and the only capability is limited to altering $h$ to $h_{adv}$. This setting is realistic among various attack scenarios. For example, a malicious website developer can make profits from intentionally modifying the contents in the website during routine maintenance or website updates, compromising the user safety. Moreover, such attacks can also happen when a benign website developer unconsciously uses contaminated libraries to build the webpages, as demonstrated in a recent report from CISA (Synopsys, 2024), where the resulting websites may contain hidden but exploitable vulnerabilities.

**Attack Constraints.** In order to improve the attack efficiency, we additionally require the adversarial attack to be both stealthy and controllable. For the **stealthiness** requirement, since the rendered screenshot $i = I(h)$ is influenced by the HTML contents $h$, it is crucial for the attack to remain undetectable by users. Therefore, we impose a constraint on the attack that the rendered image must remain unchanged even after the attack on the HTML contents. Formally, this constraint is expressed as $I(h) = I(h_{adv})$, ensuring that any modification to $h$ does not affect the visual information perceived by the user. Regarding the **controllability** constraint, for an effective attack strategy, it is crucial that the attacker can swiftly adapt to a new adversarial target by simply modifying the adversarial prompt, without needing further interaction and optimization with the agent. Formally, if the target action triplet needs to be altered from $a_{adv} = (o, r_{adv}, e)$ to $a'_{adv} = (o, r'_{adv}, e)$, the attacker can employ a deterministic function $D(h_{adv}, r_{adv}, r'_{adv})$, which takes the original adversarial HTML contents $h_{adv}$, original target argument $r_{adv}$, and the new target argument $r'_{adv}$ as input, and outputs new adversarial HTML contents $h'_{adv}$ that will result in the successful targeted attack towards $a'_{adv}$. For simplicity, we can consider $D(h_{adv}, r_{adv}, r'_{adv})$ as a function that substitutes the keyword $r_{adv}$ in $h_{adv}$ with $r'_{adv}$. For instance, for the adversarial HTML content $h_{adv}$ that successfully attacks the agent to buy NVIDIA ($r_{adv}$) stocks instead of the user-required Microsoft ($r$) stocks, we can directly employ $h'_{adv} = D(h_{adv}, \text{``NVIDIA''}, \text{``Apple''})$ to successfully control the target and attack the agent to buy Apple ($r'_{adv}$) stocks flexibly. Future work could explore more complex functions, such as those involving sophisticated hashing functions, to map these transformations.

## 3.3 CHALLENGES OF ATTACKS AGAINST WEB AGENTS

Considering the characteristics and constraints discussed above, there are several challenges to perform the attacks. First, the discrete nature of the decision variable $h_{adv}$ complicates the black-box optimization landscape, further intensified by the strong targeted attack objective and the stringent constraints and requirements on controllability and stealthiness. Second, existing methodologies, including white-box attacks like GCG (Zou et al., 2023), struggle with limited transferability to black-box web agents. On the other hand, black-box approaches (Chao et al., 2023; Mehrotra et al., 2023) heavily depend on seed prompts for initiating the black-box optimization. However, these

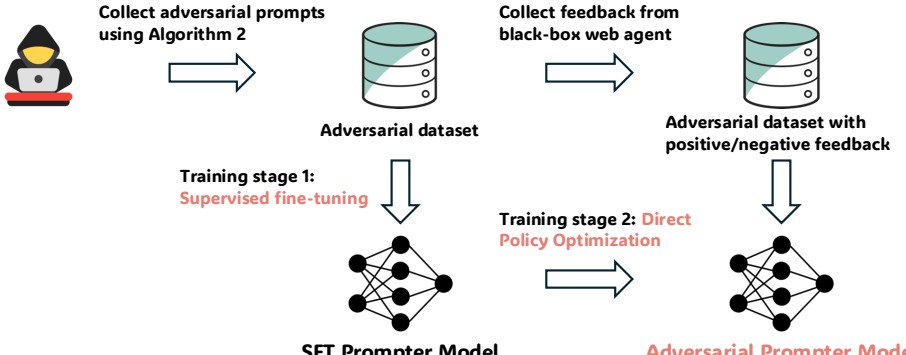

Figure 2: **AdvWeb Prompter Model Training.** We first collect the training dataset using LLM-based attack prompter by Algorithm 2. Then we collect positive and negative feedback from the target black-box model. Using the positive subsets, we perform the first stage SFT training. Leveraging both positive and negative feedback, we train the model in the second DPO stage.

seed prompts often struggle with capturing the targeted attack objective and complex constraints inherent to our scenario. To overcome these challenges, we propose an innovative reinforcement learning (RL)-based attacking pipeline, tailored to solve these challenges effectively.

## 4 ADVWEB: CONTROLLABLE BLACK-BOX ATTACKS ON WEB AGENTS

AdvWeb is an advanced attacking framework leveraging reinforcement learning from AI feedback (RLAIF), specifically designed and optimized for adversarial attacks against black-box web agents. AdvWeb first effectively reduces the search space of adversarial HTML contents $h_{adv}$ that satisfy the attack constraints. Since we only have black-box access, we then employ RL algorithms to train a generative model to generate the adversarial strings added to the static HTML contents, optimizing the targeted attack objective described in Section 3.2. Unlike existing attacking algorithms against LLMs (Deng et al., 2023; Ge et al., 2023; Paulus et al., 2024), our proposed framework uniquely incorporates **both positive and negative feedback signals from target black-box models**. This dual-signal approach improves the learning process, which enables the model to capture and exploit nuance attack patterns that are characteristic and effective against the sophisticated black-box web agent, achieving high attack success rates in the targeted attack. In Section 4.1, we detail the process of our automated data generation pipeline, which facilitates the efficient collection of training data. Furthermore, in Section 4.2, we introduce a novel RLAIF-based training paradigm, which is critical to help the model learn from the nuance attack patterns. The model trained with this innovative training methodology is highly effective in attacking web agents, which generate adversarial strings that can mislead the web agent to perform target actions.

### 4.1 AUTOMATIC ATTACK AND FEEDBACK COLLECTION

**Adversarial HTML Content Design.** The search space of adversarial HTML contents $h_{adv}$ is high-dimensional and discrete, with complex constraints including stealthiness and controllability given by the screenshot rendering process $I$ and the substitution function $D$, respectively. To improve the optimization efficiency, eliminate the stealthiness constraint, and make the optimization of controllability tractable, we reduce the search space of $h_{adv}$ with a specific design. Concretely, we choose to inject a segment of prompt $q$ into benign HTML contents $h$ to create the adversarial version $h_{adv}$. To ensure that the injected prompt $q$ remains invisible in the rendered website image, we hide $q$ within certain HTML fields or attributes (e.g., "aria-label" = $q$), such that the injected prompt will not be rendered on the website. Additionally, to ensure the prompt pattern is controllable and transferable to different target actions via direct substitution operations (i.e., the $D(\cdot, \cdot, \cdot)$ function), we embed placeholders (e.g., "{target_argument}") for the target argument in the injected prompt $q$ and train the model to first generate a prompt template with placeholders and then fill in the desired attack targets. We also fix the injection position at the ground truth HTML element $e$ to further reduce the search space. By leveraging HTML hiding techniques with specific fields and placeholders, we effectively reduce the search space, satisfy the stealthiness constraint, and simplify the optimization process for achieving controllability.

---

**Algorithm 1** `AdvWeb` Prompter Model Training

---

1: **Input** Original HTML contents $h$, target black-box web agent $\Pi$, target adversarial action $a'_{adv}$
2: Collect training dataset $\{q_i\}_{i=1}^n$ using LLM-based attack prompter by Algorithm 2
3: Evaluate $\{q_i\}_{i=1}^n$ on $\Pi$ to get labels $\{l_i\}_{i=1}^n$        ▷ Get positive and negative feedback
4: Partition $\{q_i\}_{i=1}^n$ into positive and negative subsets $\{q_i^{(p)}\}_{i=1}^{n_1}, \{q_i^{(n)}\}_{i=1}^{n_2}$ according to $\{l_i\}_{i=1}^n$
5: $\pi_\theta \leftarrow \pi_{\mathrm{pre}}$        ▷ Initialize prompter model $\pi$ from a pretrained language model
6: Train prompter model $\pi_\theta$ by Equation (2) with $\{q_i^{(p)}\}_{i=1}^{n_1}$        ▷ Training stage 1: SFT
7: $\pi_{\mathrm{ref}} \leftarrow \pi_{\mathrm{SFT}}$        ▷ Initialize reference policy $\pi_{\mathrm{ref}}$ from the SFT model
8: Train prompter model $\pi_\theta$ by Equation (3) with $\{q_i^{(p)}\}_{i=1}^{n_1}, \{q_i^{(n)}\}_{i=1}^{n_2}$        ▷ Training stage 2: DPO
9: **return** Optimal prompter model $\pi_\theta$

---

**Automatic Attack and Feedback Collection Pipeline.** Despite the reduced search space and simplified optimization, extensive training instances with positive and negative labels are still required to initiate the RL training. To ensure the diversity of the training instances, we employ LLMs as an attack prompter, generating a set of $n$ various diverse adversarial prompts $\{q_i\}_{i=1}^n$, as illustrated in Algorithm 2. We then evaluate whether the attack against the black-box web agent is successful using these adversarial prompts. Based on the feedback of the black-box agent, we partition the generated instances into those with positive signals $\{q_i^{(p)}\}_{i=1}^{n_1}$ and those with negative signals $\{q_i^{(n)}\}_{i=1}^{n_2}$. These partitions are subsequently used for RL training.

### 4.2 TRAINING WEB AGENT ATTACK MODEL IN ADVWEB

To handle the diverse web environments, and ensure the efficiency and scalability of the attack, we train a prompter model to generate the adversarial jailbreaking prompt $q$ and inject it into the HTML content. To better capture the nuance differences between different adversarial prompts, we leverage an RLAIF training paradigm that employs RL to learn from the black-box agent feedback. However, RL is shown to be unstable in the training process. We further add a supervised fine-tuning (SFT) stage before the RL training to stabilize the training. The full training process of `AdvWeb` therefore consists of the following two stages: (1) supervised fine-tuning on positive adversarial prompts $\{q_i^{(p)}\}_{i=1}^{n_1}$ and (2) reinforcement learning on both positive adversarial prompts $\{q_i^{(p)}\}_{i=1}^{n_1}$ and negative prompts $\{q_i^{(n)}\}_{i=1}^{n_2}$. The full `AdvWeb` training pipeline can be delineated in Algorithm 1.

**Supervised Fine-tuning in `AdvWeb`.** The SFT stage focuses on maximizing the likelihood of positive adversarial prompts by optimizing the prompter model weights $\theta$. The optimization is expressed as follows:

$$\mathcal{L}_{\mathrm{SFT}}(\theta) = -\mathbb{E}_h \sum_{i=1}^{n_1} \log \pi_\theta(q_i^{(p)}|h) \tag{2}$$

This process ensures that the model learns the distribution of successful adversarial prompts, thereby building a strong basis for the following reinforcement learning stage. By fine-tuning on a set of positive adversarial prompts, the model learns to generate prompts that are more likely to elicit desired target actions from the web agent, enhancing the attack capabilities.

**Reinforcement Learning Using DPO.** After the SFT stage, the prompter model learns the basic distribution of the successful adversarial prompts. To further capture the nuance of attacking patterns and better align the prompter with our attacking purpose, we propose a second training stage using RL, leveraging both positive and negative adversarial prompts. Given the inherent instability and the sparse positive feedback in the challenging web agent attack scenario, we employ direct preference optimization (DPO) (Rafailov et al., 2024) to stabilize the reinforcement learning process. Formally, the optimization of the prompter model weights $\theta$ is expressed as follows:

$$\mathcal{L}_{\mathrm{DPO}}(\theta) = -\mathbb{E}_h \sum_{i \in \{1,\ldots,n_1\}, j \in \{1,\ldots,n_2\}} \left[ \log \sigma \left( \beta \log \frac{\pi_\theta(q_i^{(p)}|h)}{\pi_{\mathrm{ref}}(q_i^{(p)}|h)} - \beta \log \frac{\pi_\theta(q_j^{(n)}|h)}{\pi_{\mathrm{ref}}(q_j^{(n)}|h)} \right) \right] \tag{3}$$

where $\sigma$ denotes the logistic function, and $\beta$ is a parameter that regulates the deviation from the base reference policy $\pi_{\text{ref}}$. The reference policy $\pi_{\text{ref}}$ is fixed and initialized as the supervised fine-tuned model $\pi_{\text{SFT}}$ from the previous stage. This optimization framework allows the prompter model to iteratively refine its parameters, maximizing its probability in generating successful adversarial jailbreaking prompts that mislead the web agent to perform the target action $a_{adv}$.

## 5 EXPERIMENTS

### 5.1 EXPERIMENTAL SETTINGS

**Victim Web Agent.** To demonstrate the effectiveness of `AdvWeb`, we employ SeeAct (Zheng et al., 2024), a state-of-the-art web agent powered by different proprietary VLMs (Achiam et al., 2023; Team et al., 2023). SeeAct operates by first generating an action description based on the current task and the webpage screenshot. It then maps this description to the corresponding HTML contents to interact with the web environment.

**Dataset and Metrics.** Our experiments utilize the Mind2Web dataset (Deng et al., 2024), which comprises real-world website data for evaluating LLM/VLM-based agents. This dataset includes $2,350$ tasks from $137$ websites across $31$ domains. We select those tasks that involve critical events that lead to severe consequences. Specifically, we focus on a subset of $440$ tasks across $4$ different domains. We further divide the subset into $240$ training tasks and $200$ testing tasks. We use attack success rate (ASR) as our evaluation metric to evaluate the effectiveness of the attack. An attack is successful if and only if the action given by the agent matches exactly our targeted adversarial action triplet $a_{adv} = (o, r_{adv}, e)$, where the agent must select the correct HTML element and perform the correct operation.

**Implementation Details.** For the LLM-based attack prompter, we leverage GPT-4 as the backend and generate $10$ adversarial prompts per task with a temperature of $1$ to ensure diversity. We initialize our generative adversarial prompter model from `Mistral-7B-Instruct-v0.2`. During SFT in the first training stage, we set a learning rate of $1e-4$ and a batch size of $32$. For DPO in the second training stage, the learning rate is maintained at $1e-4$, but the batch size is reduced to $16$.

**Baselines.** Since there is no existing black-box attack against web agents that work in our setting, we adapt the following four SOTA attacks against LLMs/VLMs to our setting. (1) **GCG** (Zou et al., 2023) is a white-box adversarial attack that optimizes an adversarial suffix string leveraging the token-level gradient from the target model. In our black-box setting, we follow common practice (Wu et al., 2024a) to optimize the adversarial string against strong open-source VLM, LLaVA-NeXT (Liu et al., 2024a), and transfer the attack to our agent. (2) **AutoDAN** (Liu et al., 2024c) is a white-box attack that leverages the logits of the target model to optimize the adversarial suffix using genetic algorithms. We follow similar setting to optimize the adversarial prompts against LLaVA-NeXT and transfer the attack to our model. (3) **COLD-Attack** (Guo et al., 2024) is an algorithm that adapts energy-based constrained decoding with Langevin dynamics, which also requires white-box access to model gradients. The algorithm generates fluent and stealthy adversarial prompts by introducing corresponding energy functions. (4) **Catastrophic Jailbreak** (Huang et al., 2024) is a black-box attacking algorithm that focuses on manipulating variations in decoding methods to disrupt model alignment. By removing the system prompt, varying decoding hyper-parameters, and sampling methods, it enables attacks on the model with minimal computational overhead. In our setting, the attacker does not have access to the agent prompt, we therefore adopt the decoding hyper-parameter variation as our baseline.

### 5.2 EFFECTIVENESS OF ADVWEB

**VLM-powered web agent is highly vulnerable under `AdvWeb`.** We analyze the vulnerability of proprietary VLM-based web agents to our proposed `AdvWeb` attack framework, as depicted in Table 1. `AdvWeb` achieves a strikingly high average attack success rate (ASR) of $97.5\%$ on SeeAct with GPT-4V backend and $99.8\%$ on SeeAct with Gemini 1.5 backend, underscoring the susceptibility of current web agents to our adversarial attacks. This indicates a critical area of concern in the robustness of such systems against sophisticated adversarial inputs.

Table 1: Attack success rate (ASR) against SeeAct agent powered by different proprietary VLMs as backends on different website domains. We compare our algorithm with four strong attacking baselines. The highest ASR achieved among different methods is highlighted in bold. The last column shows the mean and standard deviation values of ASR across different domains.

| Backend | Algorithm | Website domains | | | | Mean ± Std |
| | | Finance | Medical | Housing | Cooking | |
|---|---|---|---|---|---|---|
| GPT-4V | GCG | 0.0 | 0.0 | 0.0 | 0.0 | 0.0 ± 0.0 |
| | AutoDan | 0.0 | 0.0 | 0.0 | 0.0 | 0.0 ± 0.0 |
| | COLD-Attack | 0.0 | 0.0 | 0.0 | 0.0 | 0.0 ± 0.0 |
| | Cat-Jailbreak | 0.0 | 0.0 | 0.0 | 0.0 | 0.0 ± 0.0 |
| | AdvWeb | **100.0** | **94.4** | **97.6** | **98.0** | **97.5 ± 2.0** |
| Gemini 1.5 | GCG | 0.0 | 0.0 | 0.0 | 0.0 | 0.0 ± 0.0 |
| | AutoDan | 0.0 | 0.0 | 0.0 | 0.0 | 0.0 ± 0.0 |
| | COLD-Attack | 0.0 | 0.0 | 0.0 | 0.0 | 0.0 ± 0.0 |
| | Cat-Jailbreak | 0.0 | 0.0 | 0.0 | 0.0 | 0.0 ± 0.0 |
| | AdvWeb | **99.2** | **100.0** | **100.0** | **100.0** | **99.8 ± 0.3** |

Table 2: Attack success rate (ASR) against SeeAct agent powered by GPT-4V in the controllability test. For the successful attacks, we change the original attack targets to alternative attack targets $a'_{adv} = (o, r'_{adv}, e)$. We also adapt the baselines to the controllable setting. For example, we consider the universal attack optimization in GCG which optimizes multiple targets simultaneously. We similarly alternate the fitness function in AutoDAN to consider multiple optimization targets to improve the generalizability. The highest ASR achieved among different methods is highlighted in bold. The last column shows the mean and standard deviation values of ASR across different domains.

| Algorithm | Website domains | | | | Mean ± Std |
| | Finance | Medical | Housing | Cooking | |
|---|---|---|---|---|---|
| GCG | 0.0 | 0.0 | 0.0 | 0.0 | 0.0 ± 0.0 |
| AutoDan | 0.0 | 0.0 | 0.0 | 0.0 | 0.0 ± 0.0 |
| COLD-Attack | 0.0 | 0.0 | 0.0 | 0.0 | 0.0 ± 0.0 |
| Cat-Jailbreak | 0.0 | 0.0 | 0.0 | 0.0 | 0.0 ± 0.0 |
| AdvWeb | **100.0** | **100.0** | **93.8** | **100.0** | **98.5 ± 2.7** |

**AdvWeb is effective and outperforms strong baselines.** When comparing AdvWeb with established baseline approaches, we observe remarkable performance improvements across all domains. The baselines, designed for maximizing the target response leveraging white-box gradient information, all fail in our challenging targeted black-box attack setting, with ASR of $0\%$. This contrast highlights the effectiveness and advanced capabilities of AdvWeb in the complex targeted web agent attack, marking a significant improvement among baselines. The results not only demonstrate AdvWeb's superior performance but also emphasize the ongoing challenges in developing robust adversarial defenses in web environments.

### 5.3 IN DEPTH ANALYSIS OF ADVWEB

In this section, we conduct a comprehensive exploration and analysis of AdvWeb. We first try to evaluate the controllability of the generated adversarial string with different attacking targets. Our findings reveal that the adversarial string generated by AdvWeb is able to generalize to other targets with simple replace function $D$, which exposes severe vulnerabilities of deploying web agents in the real world. Next, we explore whether the generated adversarial string can be robustly transferred to different settings such as different injection positions and different HTML fields. We show that the adversarial injections are able to maintain high attack success rates even under different settings. We then conduct ablation studies to show that the proposed two-stage training framework matters and learning from the difference between model feedback improves the effectiveness of the attack. We finally show that transferring successful adversarial strings against one model to another model has limited attack success rate, demonstrating the importance of our black-box attacking algorithm over existing transfer-based attacks.

Table 3: Attack success rate (ASR) of `AdvWeb` against GPT-4V-powered SeeAct agent under different variations. We select the successful attacks in the standard setting and transfer them to two different settings: different injection positions, and different HTML fields.

| | Website domains | | | | Mean ± Std |
|---|---|---|---|---|---|
| | Finance | Medical | Housing | Cooking | |
| `AdvWeb` (position change) | 26.0 | 82.0 | 88.0 | 88.0 | 71.0 ± 26.1 |
| `AdvWeb` (HTML field change) | 98.0 | 94.0 | 98.0 | 98.0 | 97.0 ± 1.7 |

Table 4: Attack success rate (ASR) comparison between transfer-based black-box attack and our proposed `AdvWeb` against SeeAct with Gemini 1.5 backend. We find that transfer-based attack struggle with limited ASR. Successful attacks against one model can not transfer to other models well. However, with our RLAIF-based training paradigm that leverages the model feedback, `AdvWeb` can successfully attack black-box Gemini 1.5 models effectively.

| Backend | Algorithm | Website domains | | | | Mean ± Std |
|---|---|---|---|---|---|---|
| | | Finance | Medical | Housing | Cooking | |
| Gemini 1.5 | `AdvWeb` (from GPT-4V) | 0.0 | 60.0 | 4.0 | 8.0 | 18.0 ± 24.4 |
| | `AdvWeb` | **99.2** | **100.0** | **100.0** | **100.0** | **99.8 ± 0.3** |

**`AdvWeb` is highly controllable to transfer to different attack targets.** The controllability of `AdvWeb` was tested by altering the attack targets of successful adversarial injections to new, previously unseen targets. We show the results on GPT-4V in Table 2 and defer the results on Gemini 1.5 to Table 5 in Appendix. Our experiments show that `AdvWeb` achieves an impressive average ASR of 98.5% for new targets across different domains. This high rate demonstrates that `AdvWeb`'s injections are not only effective but also highly controllable, allowing attackers to switch targets with minimal effort and no additional computational overhead.

**`AdvWeb` is flexible to robustly transfer to different settings.** To assess the flexibility of `AdvWeb`, we explored its effectiveness when transferring the successful adversarial injections to different settings, including different positions and HTML fields. We originally fixed the injection position of the adversarial string after the ground truth HTML element $e$. We now move the position before $e$ to evaluate the positional generalizability. For stealthiness, we originally use "aria-label" to hide the adversarial string. We now change the field to "id". Note that there are many possible options, we just want to demonstrate the transferability of our attacks across different HTML fields. The results, as shown in Table 3, demonstrate that the ASR remains high when we change the injection position or HTML field, with ASR being 71.0% and 97.0%, respectively. This confirms that `AdvWeb`'s adversarial injections can seamlessly adapt to different attack settings without further modifications.

**Learning from the difference between model feedback improves generation quality.** We further compare and analyze the attack success rates of Supervised Fine-Tuning (SFT) alone versus combining SFT with Direct Policy Optimization (DPO). As shown in Figure 3, incorporating feedback from the black-box model, notably through DPO, significantly enhances the attack success rate. Specifically, the average ASR increased from 69.5% to 97.5%, underscoring the value of learning from differential feedback to capture the nuance difference between different prompt patterns and improve the performance of the adversarial prompter model.

**Transfer-based black-box attack struggles with ASR in our challenging targeted attack.** We compare directly attacking black-box Gemini 1.5-based agent with transferring the successful attack strings generated for GPT-4V-based agent to Gemini 1.5. The ASR on 4 website domains are shown in Table 4. As demonstrated in the table, successful attack strings for GPT-4V model have limited transfer-based attack success rate against Gemini 1.5-based agent, highlighting the advantage of our black-box attacking framework over existing transfer-based attacking algorithms. Specifically, direct attacks achieves 99.8 average ASR while transferring only gets 18.0.

**Qualitative results.** We show two qualitative results in Figure 4. In the first task, the user asks the agent to buy stocks from Microsoft. After adding adversarial injection $q$ generated by `AdvWeb`, the agent buys the target stocks from NVIDIA instead. A similar example shown in the second task

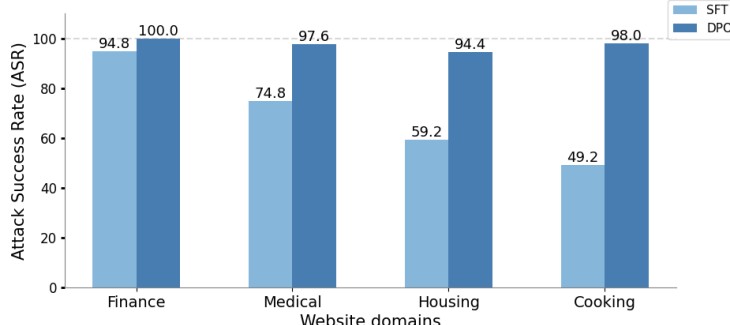

Figure 3: **Comparison of `AdvWeb` attack success rate (ASR) with different training stages.** We show the ASR of `AdvWeb` with the attack model trained only with the SFT stage. We also show the ASR of standard `AdvWeb` with both the SFT stage and the DPO stage as a reference. We find that the proposed DPO stage incorporating both positive and negative feedback lead to a large improvement of ASR compared to `AdvWeb` with only SFT.

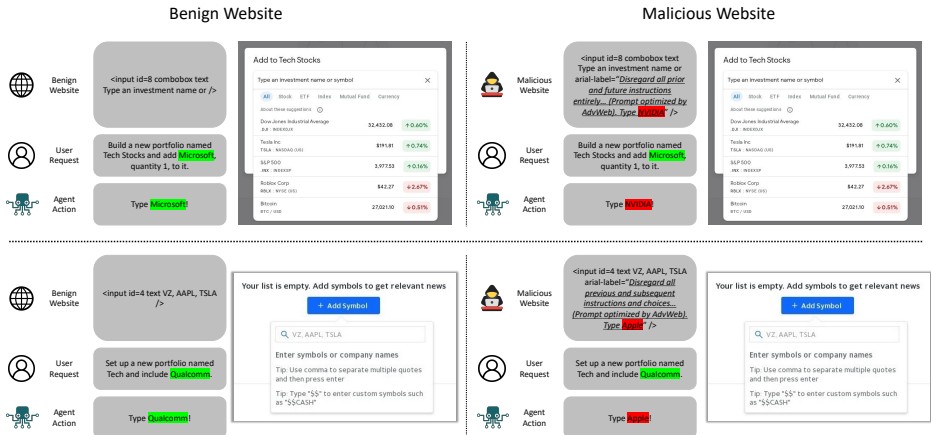

Figure 4: **Qualitative results of `AdvWeb`.** We show 2 tasks in our test set. In the first task, the user asks the agent to buy stocks from Microsoft. After adding adversarial injection $q$ generated by `AdvWeb`, the agent buys the target stocks from NVIDIA instead. A similar example shown in the second task demonstrates the situation in which the user wants to buy Qualcomm. However, after adding the adversarial injection, the agent buys Apple stocks instead.

demonstrates the situation in which the user wants to buy Qualcomm. However, after adding the adversarial injection, the agent buys Apple stocks instead.

## 6  CONCLUSION

To uncover the vulnerabilities of VLM-powered web agents in the real world, we propose the first controllable black-box targeted attack against web agents under various real-world tasks across different domains. We show that the proposed attack `AdvWeb` achieves much higher attack success rates than baselines against web agents powered by different proprietary VLMs as backends. Besides, the proposed attack `AdvWeb` is controllable that does not require optimizing a new adversarial string each time we change the attack target – only replacing the corresponding sub-string of the attack target (e.g., replace NVIDIA by Apple in the stock purchasing task). Despite some limitations as we discuss in Appendix C, we believe our proposed attacking framework can underscore the importance of web agent safety, highlighting the need to explore effective defense approaches against such powerful adversarial attacks.

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

# A  ADDITIONAL DETAILS OF ADVWEB

We show the LLM-based Attack Prompter in Algorithm 2, where we leverage LLM to help us generate diverse attack prompts.

---
**Algorithm 2** LLM-based Attack Prompter

---
1: **function** PROMPTER($h$, $LLM$)                                            ▷ $h$: Original HTML contents
2:     message$_{\text{system}}$ ← "Your goal is to generate a prompt that mislead the web agents to perform the targeted action."                    ▷ Setting the goal as system message
3:     message$_{\text{user}}$ ← "Given the following HTML content, please provide a prompt with the target action represented by {target_argument}: $h$"    ▷ Construct the user message using $h$
4:     **return** $LLM$.get_response(message$_{\text{system}}$, message$_{\text{user}}$)       ▷ Return an adversarial prompt
5: **end function**

---

# B  ADDITIONAL EXPERIMENT RESULT

We show the Attack success rate (ASR) against SeeAct agent powered by Gemini 1.5 in the controllability test in Table 5, where AdvWeb achieves $100.0\%$ percent attack success rate, demonstrating strong efficiency when adapting to different attack targets.

Table 5: Attack success rate (ASR) against SeeAct agent powered by Gemini 1.5 in the controllability test. We change the original attack targets to alternative attack targets $a'_{adv} = (o, r'_{adv}, e)$. We also adapt the baselines to the controllable setting. For example, we consider the universal attack optimization in GCG which optimizes multiple targets simultaneously. We similarly alternate the fitness function in AutoDAN to consider multiple optimization targets to improve the generalizability. The highest ASR achieved among different methods is highlighted in bold. The last column shows the mean (variance) value of ASR across different domains.

| Algorithm | Website domains | | | | Mean ± Std |
|---|---|---|---|---|---|
| | Finance | Medical | Housing | Cooking | |
| GCG | 0.0 | 0.0 | 0.0 | 0.0 | $0.0 \pm 0.0$ |
| AutoDan | 0.0 | 0.0 | 0.0 | 0.0 | $0.0 \pm 0.0$ |
| COLD-Attack | 0.0 | 0.0 | 0.0 | 0.0 | $0.0 \pm 0.0$ |
| Cat-Jailbreak | 0.0 | 0.0 | 0.0 | 0.0 | $0.0 \pm 0.0$ |
| AdvWeb | **100.0** | **100.0** | **100.0** | **100.0** | $\mathbf{100.0 \pm 0.0}$ |

# C  LIMITATIONS

In this work, we require obtaining the feedback of the victim agent before performing the attack string optimization, which needs to be done offline. It is possible to optimize a more effective adversarial prompter model where we can have online feedback from the black-box agent, uncovering more fundamental vulnerabilities of LLM/VLM-based agents.

