# OpenReview forum: "AdvWeb: Controllable Black-box Attacks on VLM-powered Web Agents"
_ICLR.cc/2025/Conference — Submitted to ICLR 2025_

### Official Review · Reviewer_7hWf · 2024-10-31

**Soundness:** 2
**Presentation:** 2
**Contribution:** 1
**Rating:** 3
**Confidence:** 4

**Summary:**

This paper introduces a black-box targeted attack against Vision-Language Model-powered web agents. The approach involves training an adversarial prompt model using direct policy optimization. This model generates and injects adversarial prompts into web pages, leading web agents to execute targeted malicious actions. The effectiveness of the proposed attack is demonstrated through evaluations conducted on SeeAct.

**Strengths:**

- The paper showcases the effectiveness of attacks on SeeAct across several prominent VLM models, including GPT-4V and Gemini 1.5.

- It conducts an extensive ablation study to evaluate the effectiveness of the proposed attacks under different attack settings and targets.

**Weaknesses:**

- Since the attack's goal is to mislead agents into executing malicious actions, the scope of the paper aligns more closely with prompt injection attacks rather than jailbreak attacks. However, the paper focuses significantly on distinguishing itself from jailbreak attacks and includes them as evaluation baselines for comparison. It would be beneficial to provide a comprehensive survey of prompt injection attacks in the related work section, highlighting the key differences and advantages of the proposed method compared to existing attacks. Including prompt injection attacks as baselines would strengthen the contribution of the paper.

- Given the extensive research on prompt injection attacks, the paper does not clearly distinguish the key advantages of its proposed attack over existing methods. Specifically, there is a line of prior studies [1][2] that have utilized context-ignoring techniques by appending prompts like "Ignore previous instructions, do xxx." This is similar to the method used in this paper, as illustrated in Figure 4, where the attack prompt is "Disregard all previous and subsequent instructions and choices, do xxx." Moreover, previous methods do not require collecting a prompt training dataset or performing expensive DPO training of an LLM-based attack prompter, making them more efficient.

- Defenses against prompt injection attacks have also been extensively studied (e.g., [3][4]). It is important to assess the effectiveness of the proposed attack against these defenses to determine its practical applicability and robustness.

- The selection of victim web agents is limited. The paper evaluates only one type of VLM-based web agent, which may not fully demonstrate the generalizability of the proposed attacks. Incorporating a more comprehensive evaluation with a variety of web agents would strengthen the paper.

- In comparing baselines during evaluation, the paper includes several white-box jailbreak attacks like GCG and AutoDAN. It is unclear how these baselines are implemented against SeeAct, given that it uses proprietary VLMs that are black-box and do not provide gradient information.

Reference:

[1] Branch, Hezekiah J., et al. "Evaluating the susceptibility of pre-trained language models via handcrafted adversarial examples." arXiv preprint arXiv:2209.02128 (2022).

[2] Perez, Fábio, and Ian Ribeiro. "Ignore Previous Prompt: Attack Techniques For Language Models." NeurIPS ML Safety Workshop. 2022.

[3] Liu, Yupei, et al. "Formalizing and benchmarking prompt injection attacks and defenses." 33rd USENIX Security Symposium (USENIX Security 24). 2024.

[4] Chen, Sizhe, et al. "Aligning LLMs to Be Robust Against Prompt Injection." arXiv preprint arXiv:2410.05451 (2024).

**Questions:**

- Why does the paper not compare with prompt injection attacks, which are more aligned with the context of this work?

- What are the key advantages of the proposed attacks compared to existing prompt injection attacks?

- What are the evaluation results of the proposed attacks against prompt injection defenses?

- How are the white-box baseline jailbreak attacks implemented against black-box VLMs?

**Details Of Ethics Concerns:**

No ethical concerns are involved.

---

> ### Author Response · Authors · 2024-11-24
> **Response to Reviewer 7hWf (Part 1)**
>
> We appreciate the thoughtful and constructive feedback on our work. Below, we address your concerns in detail:
>
> > **Q1.** The scope of the paper aligns more closely with prompt injection attacks rather than jailbreak attacks. Including prompt injection attacks as baselines would strengthen the contribution of the paper.
>
> Thank you for the insightful suggestion. Following your advice, we have incorporated additional baselines for comparison. Specifically, we evaluated the attack success rates of (1) heuristic-based human-written adversarial strings, (2) GPT-generated adversarial strings obtained from Algorithm 2, (3) SFT, and (4) our proposed AdvWeb framework (SFT + DPO). The results show that human-written and GPT-generated strings generally achieve lower attack success rates (ASR) compared to SFT, except in the cooking domain, likely due to the simpler nature of tasks in that domain. Notably, our AdvWeb framework consistently achieves the highest ASR across all domains, demonstrating the effectiveness of combining SFT and DPO in generating adversarial strings.
>
> | Methods       | Finance | Medical | Housing | Cooking |
> |---------------|---------|---------|---------|---------|
> | Human written | 16.0    | 8.0     | 16.0    | 72.0    |
> | GPT generated | 63.6    | 67.3    | 19.6    | 96.7    |
> | SFT           | 94.8    | 74.8    | 59.2    | 49.2    |
> | AdvWeb     | 100.0   | 94.4    | 97.6    | 98.0    |
>
> These results highlight the superior performance of AdvWeb, particularly in complex domains like finance and medical, where the combination of SFT and DPO significantly boosts the attack's effectiveness.
>
> > **Q2.** The paper does not clearly distinguish the key advantages of its proposed attack over existing methods.
>
> The key advantages of our method over traditional prompt injection attacks include:
>
> - Automation and Scalability: While previous approaches rely on handcrafted adversarial prompts, our method employs automated training of an attack model using DPO, significantly reducing manual effort and enabling scalability.
>
> - Optimized Attack Generation: Unlike traditional methods that rely on handcrafted prompts, our framework uses DPO to systematically generate attack strings. This ensures higher success rates by tailoring prompts to specific agent behaviors through iterative optimization.
>
> > **Q3.** Effectiveness against injection defenses.
>
> Thank you for highlighting this important point. Existing defenses against prompt injection attacks, such as adversarial training, input sanitization, and validation (e.g., ignoring hidden HTML attributes like aria-label or relying solely on visible user-intended elements), can mitigate such attacks. We will include an evaluation and discussion in the revised paper.
>
> > **Q4.** The paper evaluates only one type of VLM-based web agent, which may not fully demonstrate the generalizability of the proposed attacks.
>
> We appreciate the reviewer’s observation regarding the evaluation scope. In our work, we chose SeeAct as the primary target agent because it is a state-of-the-art (SOTA) web agent powered by advanced Vision-Language Models (VLMs) such as GPT-4V and Gemini 1.5. SeeAct is widely recognized in the community for its ability to perform complex, multimodal tasks on real-world websites, making it an appropriate benchmark for assessing the practical effectiveness of our attack framework.
>
> In future work, we aim to systematically test our approach on a wider variety of VLM-powered web agents and datasets. We will also discuss these limitations and our planned extensions more explicitly in the revised version of the paper. Thank you for highlighting this important direction for improvement.

---

> ### Author Response · Authors · 2024-11-24
> **Response to Reviewer 7hWf (Part 2)**
>
> > **Q5.** It is unclear how these baselines are implemented against SeeAct, given that it uses proprietary VLMs that are black-box and do not provide gradient information.
>
> Thank you for the question. Since there is no existing black-box attack against web agents in our setting, we adapted the following attacks to our targeted black-box attack scenario:
>
> 1. GCG: This white-box attack optimizes adversarial suffix strings using token-level gradients from the target model. In our setting, where gradient access to the black-box agent is unavailable, we optimized adversarial strings against a strong open-source VLM, LLaVA-NeXT, and transferred them to the GPT-4V-based web agent.
>
> 1. AutoDAN: AutoDAN: This white-box attack uses genetic algorithms to optimize adversarial suffixes by leveraging logits from the target model. We optimized adversarial prompts against LLaVA-NeXT and transferred them to the black-box agent.
>
> 1. COLD-Attack: This method generates stealthy adversarial prompts by leveraging energy-based constrained decoding, requiring access to model gradients. Similarly, we optimized adversarial strings on LLaVA-NeXT and transferred them.
>
> 1. Catastrophic Jailbreak: This black-box attack manipulates decoding variations to disrupt alignment, focusing on system prompts and decoding parameters. Since our setting does not allow access to system prompts, we adapted the decoding parameter variation.
>
> The poor performance of these baselines can be attributed to two key factors. First, the optimization goal in our attack is different from prior works, which typically aim to elicit affirmative responses or bypass filters. In contrast, our attack requires preserving the original response while altering the target element or action within it, making it inherently more complex. Second, adversarial strings optimized against white-box models like LLaVA-NeXT are challenging to transfer effectively to proprietary black-box models like GPT-4V, especially in targeted attack settings.
>
> We will clarify these adaptations and the unique challenges of our setting in the revised paper.

---

> > ### Comment · Reviewer_7hWf · 2024-12-02
> >
> > Thank you for your response and the additional experiments. After reviewing the rebuttal, I still have some questions. Since the paper is closely tied to the formulation of prompt injection attacks, it is important to evaluate the attack's effectiveness against state-of-the-art defenses to validate its robustness. Additionally, evaluating a single target web agent is insufficient to demonstrate the generalizability of the attack. Furthermore, the comparison with baselines appears somewhat unfair, as most baselines were originally proposed for a white-box setting. The adaptation through transferability, as implemented by the authors, may have negatively impacted their performance.

---

### Official Review · Reviewer_d1HB · 2024-11-02

**Soundness:** 3
**Presentation:** 3
**Contribution:** 3
**Rating:** 5
**Confidence:** 5

**Summary:**

This paper introduces a novel black-box attack framework, AdvWeb, targeting website agents driven by Visual Language Models (VLMs). Under the black-box framework, AdvWeb proposes a two-stage training paradigm. First, it conducts supervised fine-tuning on positive adversarial prompts to obtain the SFT Prompter Model. Subsequently, it leverages the black-box feedback from the target agent combined with DPO strategy for reinforcement learning to train an adversarial prompt model, generating and injecting adversarial prompts into web pages to induce specific adversarial behaviors from the network agent. This method is characterized by its stealth and controllability, with a high success rate of attacks.

**Strengths:**

- The paper presents AdvWeb, the first black-box target attack framework for VLM-based website agents, proposing a method to train adversarial prompt models through reinforcement learning with DPO in a black-box setting, which is innovative and effective.

- The method ensures stealth, controllability, and black-box scenarios, which adds greater practical value.

- The experimental results demonstrate the effectiveness of AdvWeb in attacking different VLM-based website agents and tasks, which helps to raise awareness in the field for developing more reliable web agents and implementing effective defense measures.

**Weaknesses:**

- The threat model considered in this paper may be impractical.

- This paper does not propose any potential defense mechanisms.

- The types of victim web agents considered are limited.

**Questions:**

Thanks for submitting your paper to ICLR 2025.

With the advancement in the intelligence of large language models, using VLM agents to automate a series of tasks is emerging as a future development trend. However, the associated security risks remain unexplored. This paper introduces AdvWeb in an attempt to expose the security risks involved in VLM-powered web agents. The topic discussed in this work is both timely and highly important.

However, I have the following concerns:

- First, regarding the threat model, I am curious about the identity of the adversary here. The adversary’s method involves embedding malicious strings in web content. Generally, if users choose commands that access official websites, how could an adversary manipulate an official site? If it is not an official website, how would the adversary ensure that their site gets indexed?

- The threat highlighted in this paper is urgent, making it essential to consider corresponding defense mechanisms. However, this paper does not discuss any defense strategies.

- The AdvWeb attack scenario, which targets web content, appears may similar to prompt injection attacks [R1]. Analyzing and comparing the differences between these two attack types would be beneficial.

- This paper selects SeeAct as the web agent framework and uses GPT-4V and Gemini 1.5 as the underlying VLMs. I am curious whether current open-source VLMs also support SeeAct. Additionally, in future edge applications, where privacy is a priority, smartphones may deploy smaller-scale VLMs locally. Analyzing the attack’s effectiveness on open-source VLMs would help demonstrate the generalizability of the attack.

- In practical applications involving user payment actions, a secondary authentication of the user’s identity is typically required, providing the user with an opportunity to review the final outcome and potentially prevent malicious actions. Does this scenario indirectly suggest that the stealthiness of the attack may be limited?

Reference

[R1] Adversarial Search Engine Optimization for Large Language Models (https://arxiv.org/abs/2406.18382)

---

> ### Author Response · Authors · 2024-11-24
> **Response to Reviewer d1HB**
>
> We appreciate the thoughtful and constructive feedback on our work. Below, we address your concerns in detail:
>
> > **Q1.** The threat model considered in this paper may be impractical.
>
> Thank you for highlighting this important aspect of the threat model. Our adversarial framework assumes scenarios where malicious strings can be embedded into web content, which can happen in several realistic cases:
>
> - Visiting Malicious Websites: Users might interact with unverified or malicious websites while conducting general web searches or browsing. These sites could be intentionally designed to exploit web agent vulnerabilities.
>
> - Compromised Trusted Websites: Even legitimate websites can occasionally fall prey to security breaches. For example, attackers could exploit weaknesses in website plugins, advertisements, or third-party integrations to inject malicious code.
>
> - Compromised Open-Source Libraries: Many websites rely on open-source packages for front-end development, which might inadvertently include malicious components. This risk is well-documented in supply chain security literature.
>
> > **Q2.** This paper does not propose any potential defense mechanisms.
>
> Thank you for raising this important question. While designing robust defenses against adversarial attacks is a non-trivial challenge, we propose the following suggestions to enhance the security of web agents:
>
> 1. Adversarial Robustness through Training: Incorporating adversarial training methods, where web agents are trained on adversarially perturbed examples, can help improve their robustness against such attacks.
>
> 1. Input Sanitization and Validation: Web agents should implement strict sanitization processes to ignore hidden HTML attributes (e.g., aria-label, id) and rely solely on visible and user-intended elements for decision-making.
>
> 1. Action Validation with Contextual Awareness: Introducing a secondary verification mechanism for critical actions, such as cross-checking proposed actions against the user’s task context, can help prevent unintended or harmful behaviors.
>
> 1. Behavioral Monitoring and Logging: Developing monitoring tools to track and analyze web agent behaviors in real time can help detect anomalies and suspicious actions, enabling timely intervention.
>
> We will include a dedicated discussion on these potential defenses in the revised paper to provide further insights into safeguarding web agents.
>
> > **Q3.** The AdvWeb attack scenario, which targets web content, appears similar to prompt injection attacks. Analyzing and comparing the differences between these two attack types would be beneficial.
>
> Thank you for this suggestion. While our method and traditional prompt injection attacks share some similarities, there are key differences:
>
> - Automation and Scalability: While previous approaches rely on handcrafted adversarial prompts, our method employs automated training of an attack model using DPO, significantly reducing manual effort and enabling scalability.
>
> - Optimized Attack Generation: Unlike traditional methods that rely on handcrafted prompts, our framework uses DPO to systematically generate attack strings. This ensures higher success rates by tailoring prompts to specific agent behaviors through iterative optimization.
>
> We will include a detailed analysis comparing AdvWeb to traditional prompt injection attacks in the related work section of the revised paper.
>
> > **Q4.** Does SeeAct support current open-source VLMs, and how does this affect the attack’s generalizability?
>
> Open-source VLMs, such as Llava, do support SeeAct as their underlying model. However, their performance is significantly limited compared to proprietary models like GPT-4V. Specifically, open-source VLMs often fail to successfully complete critical steps in web-based tasks. For this reason, we focus our analysis on cases where the agent successfully completes the original task, as attacking scenarios where the task itself fails hold limited significance. We will further discuss these limitations in the revised paper.
>
> > **Q5.** In practical applications involving user payment actions, secondary authentication may limit the stealthiness of the attack. How does this scenario affect your evaluation?
>
> Our evaluation focuses on the step-based attack success rate (ASR), which measures the success of adversarial attacks at individual decision-making steps within a task. This granular approach is necessary due to the current limitations of web agents, which often exhibit relatively low end-to-end task completion rates. While step-level evaluations provide valuable insights into specific vulnerabilities and attack effectiveness, they may not fully capture the cumulative risks associated with an agent’s ability to complete entire user requests. We will add corresponding discussion in the limitation section.

---

### Official Review · Reviewer_Z2a9 · 2024-11-02

**Soundness:** 1
**Presentation:** 1
**Contribution:** 3
**Rating:** 3
**Confidence:** 4

**Summary:**

This paper introduces an attack framwork against generalist web agents that parse websites through screenshots and HTML. The authors attempt to demonstrate how a malicious web-developer (either in-house or via open-source HTML that other devs blindly import) could use HTML that’s invisible to human users to influence the actions of a generalist web agent.

They optimise attacks in a black box setting against models based on GPT-4V and Gemini 1.5, scaffolded as web agents through the SeeAct framework. They generate attacks using GPT-4 to generate HTML with hidden instructions, then optimise an adversarial prompter against the target model using RLAIF so that those hidden instructions are maximally effective against the target.

The SeeAct framework allows the victim models to read HTML elements in the webpage that a normal human user wouldn't normally see. They inject adversarial prompts into these hidden (the authors call these "invisible") HTML fields, and their results show success in influencing SeeAct agents to take different actions than their user prompt instructed them: for example, when told to purchase MSFT stocks on a financial website, the injected HTML prompt misleads the agent to buy e.g. Nvidia instead.

The authors display high success rate against the target models, especially in the face of their four baselines, all of which achieve 0% ASR across every domain, on both models. Three of their baselines (GCG, AutoDan, COLD-Attack) involve optimising adversarial prompts against whitebox models and hoping those transfer, and the other, Catastrophic Jailbreak, requires no access to model internals but control over the model's decoding hyperparameters.

In addition to requiring that their injected prompts are "stealthy" - i.e. invisible to a normal user, the authors also emphasise the controllability of their attacks - i.e. the ease with which attackers can switch out one set of injected instructions for others. With this property, they also demonstrate that their attack strings can be adapted to transfer between different task-domains.

The authors also report that their attacks transfer well between the two models they attack, and between different HTML fields (using the same successful prompts), which suggests that their attacks aren't brittle to the specifics of the model / HTML field they were optimised for.

**Strengths:**

I believe what the authors are trying to achieve is novel, interesting, and important: they target a timely and significant concern that will only become more prevalent as generalist web agents are deployed and relied on.

The work also seems novel. My impression from the literature is that there aren't many other papers showing successful attacks/injections against web agents, especially black-box, and via means that would be invisible to most humans.

The authors' constraints of 1.) only allowing themselves black-box access to the target models, and 2.) their constraint of "stealthiness" increase the difficulty and realism of their attack framework. Attackers hiding malicious instructions in hidden HTML fields seems realistic and aligns with their threat model.

Given how realistic their attacks seem, and how high their reported ASRs are, this paper could promote increased safeguards on generalist web agents - or in the least could present a compelling demonstration that users should be careful when selecting which agent frameworks to rely on.

It's impressive that their set their success threshold quite high - at achieving exactly the the action their attack targets (instead of e.g. just distracting the agent to make it fail at its main instruction).

**Weaknesses:**

While the paper presents a novel and important problem setting, I don't feel confident that the paper provides me the information I'd need to evaluate if the attacks it introduces are as successful and significant as the authors claim. I think it's quite possible the authors have all the data they need to convince me otherwise, but I urge them to include it in the paper.

Overall, reading the paper gives me a weak sense of what generalist web agents can be influenced to do, or how significant the negative outcomes of these attacks could be. I am not able to evaluate the risks of their attack method if all I know is that they cause models to take unspecified targeted actions in a domain the paper only labels as "cooking" or "housing." If the authors could provide many more examples of successful tasks & attacks, and their consequences - at least one for each attack domain, their claims of significance would be much more convincing.

In their related work, the authors claim that they beat "strong baselines." But I'm unconvinced that any baseline on a task that achieves 0% ASR on every task, on both models they test, should be considered a strong baseline. The authors claim that there are no analogous black-box attacks in this setting, which I can't refute from what I've read elsewhere. However, I'm confused why they don't compare at all against the methodologies from the papers they list in the existing body of research from "Existing Attacks against Web Agents." Even if those methods are manual, where this paper's methods are learned, it feels like a much more informative comparison. I would urge that the authors find a more successful baseline to test against, and ideally show some results comparing their success rate to at least one other method for steering the same web-based agents.

Because of the weakness of the baselines they compare against, and the lack of comparison to other methods for achieving the same ends. I'm left without much context to evaluate how powerful this method is. I recognise that the attack domain is novel, and I found figure 3 helpful for understanding that their training pipeline helps their models achieve higher success after DPO. Discussion of more ablations, and in particular considerations of alternative methods for training and optimising their attacks, and why they weren't as successful, would be informative to my understanding of this paper.

I don't leave the paper with a strong sense of the offense-defense balance in this setting. In particular, the paper might benefit from more detail on how expensive it is to generate these attacks, as the authors do not provide much detail on how expensive the training process is (in steps or dollars) for their RLAIF pipeline outlined in Algorithm 1. For any attack, it seems important to know how quickly and cheaply attackers could generate new attacks when the victim models are updated. Further, if their RLAIF pipeline required very many training steps, it's plausible that the developers of these web agents could become aware. I would be interested to see e.g. how the ASR of their framework increases throughout training.

I'm confused why the title of the paper focuses on attacking "VLM-Powered Web Agents." While true - the SeeAct framework only employs VLMs - as far as I can tell nothing about the victims' multimodality is being attacked, simply their parsing of HTML text. My first impressions of the paper led me to expect exploits against the multimodality of these models, which was ultimately incorrect. I suggest the authors remove "VLM powered" from the paper title.

The authors repeatedly stress the importance of the controllability of their HTML attacks (including in the title). That is, that the same attack strings can be easily adapted to cause different target actions on the same task. Any examples of the ways in which their HTML attack strings are editable would be helpful. But more importantly I do not get a sense from the paper why this is important under their threat model. I think the answer may be a question of cost: that it is cheap to retool successful attack strings for a different purpose, but the existing wording in the paper does not mention this. The most I see is that the authors claim: "allowing attackers to switch targets with minimal effort and no additional computational overhead." This remark is on the penultimate page of the paper, which I think is too late and too little justification for what is introduced as a key constraint of their attacks - even in the title of the paper. More commentary on cost, as I request above, would also have made the relevance of this constraint - which seems technically sound - much clearer.

The transfer results aren't as convincing as their main results - especially since the ASR is quite varied across different domains, achieving 0% transfer on probably the most compelling domain for their threat model (online finance tasks).

Appendix C, addressing limitations is two sentences long and claims that "It is possible to optimize a more effective adversarial prompter model." The authors don't expand on this claim and I would rather they address more of the limitations in their threat model that this review (& if relevant, others) highlights.

Some weakness in writing & setting that ought to be addressed but should be trivial to fix:

The authors refer to Algorithm 2 throughout the paper including in the second figure and in their description of Algorithm . Algorithm 2 can in fact be found as the only entry in Appendix A, while Algorithm 1, which in my opinion requires Algorithm 2 to understand it, is in the main body. Algorithm 2 is critical for understanding how they generate the initial malicious HTML requests that they then label for RLAIF using the victim model, and I was confused at first because the authors didn't list where algorithm 2 could be found.

Some claims in the abstract and introduction seem too strong for the level of evidence this paper provides. For example:
* That Vlms “have revolutionized the creation of generalist web agents … thereby boosting human efficiency and productivity” requires at least some citation or evidence
* That their choice of injecting into unseen HTML fields makes it “nearly impossible for users to detect tampering” feels like a stretch: can't any user inspect the page's HTML? I appreciate that most users wouldn't.

There's also an error on the final page, where a paragraph break interrups a sentence immediately before the conclusion, leaving a hovering sentence fragment that reads "[newline] demonstrates the situation in which the user wants to buy Qualcomm. However, after adding the adversarial injection, the agent buys Apple stocks instead." This sentence fragment, which appears to be an incomplete draft of the final setence of Section 5, needs to be addressed before the paper can be acceptable.

**Questions:**

1. What other baselines did you consider, and what would it look like for you to compare your method to the attacks against web agents that you list in related work?
2. How expensive was the training & optimisation process for generating successful attacks?
2. Could you explain what signal you optimise against from the target model - does the victim model refuse?
3. When you inspect samples maually, do you get a sense of why transfer from GPT-4V to Gemini on finance tasks so much lower?
4. How did you select the tasks from Mind2Web to test against? Was it manual inspection? What were your criteria to judge that a task involves "critical events that lead to severe consequences"?

---

> ### Author Response · Authors · 2024-11-24
> **Response to Reviewer Z2a9 (Part 1)**
>
> We thank Reviewer Z2a9 for their detailed and constructive feedback. Below, we address your questions and concerns:
>
> > **Q1.** I am not able to evaluate the risks of their attack method if all I know is that they cause models to take unspecified targeted actions in a domain the paper only labels as "cooking" or "housing." If the authors could provide many more examples of successful tasks & attacks, and their consequences - at least one for each attack domain, their claims of significance would be much more convincing.
>
> In the housing domain, the potential economic loss is significant, as tasks often involve selecting properties where price and location are critical factors. We selected these domains such as "cooking" and "housing" as representative examples to demonstrate the diversity and adaptability of our attack framework. These examples primarily serve to illustrate the framework's ability to handle diverse domains and adapt to other sensitive applications, such as finance or healthcare, where the stakes and risks can be much higher. By focusing on diversity, we highlight the broad applicability and flexibility of our method, rather than its limitations to specific domains.
>
> > **Q2.** baseline on a task that achieves 0% ASR on every task
>
> We appreciate this important question. Below, we provide details on the baselines and the challenges they face in achieving successful attacks in our black-box setting:
>
> - GCG: This white-box attack optimizes adversarial suffix strings using token-level gradients from the target model. In our black-box setting, where gradient access is unavailable, we adapted GCG by optimizing adversarial strings against a strong open-source VLM, LLaVA-NeXT, and transferring them to the GPT-4V-based web agent.
>
> - AutoDAN: AutoDAN uses genetic algorithms to optimize adversarial suffixes based on logits from the target model. We similarly used LLaVA-NeXT to generate adversarial prompts and transferred them to the black-box GPT-4V agent.
>
> - COLD-Attack: This method uses energy-based constrained decoding for stealthy adversarial prompt generation, relying on model gradients. We optimized adversarial strings on LLaVA-NeXT and transferred them to GPT-4V.
>
> - Catastrophic Jailbreak: This black-box attack disrupts alignment by manipulating decoding variations, focusing on system prompts and decoding parameters. Since our setting restricts access to system prompts, we adapted decoding parameter variations for our experiments.
>
> Despite significant efforts to adapt these methods, the baselines achieved 0% ASR due to the following reasons:
>
> 1. Different Optimization Objectives:
> Most prior attacks, including the baselines, focus on simpler objectives, such as bypassing filters or eliciting affirmative responses. In contrast, our attack framework requires preserving the original response structure while altering specific target elements or actions, introducing a dual-layer complexity to the optimization process. The baselines are not equipped to handle these more stringent requirements.
>
> 1. Limited Transferability:
> Adversarial strings optimized against white-box models, such as LLaVA-NeXT, exhibit poor transferability to proprietary black-box models like GPT-4V. This issue is especially pronounced in targeted attack scenarios, where high precision is needed to alter specific actions or elements within the agent's response.
>
> Our method, AdvWeb, directly addresses these limitations by incorporating reinforcement learning with feedback from the black-box agent, ensuring both high transferability and targeted precision. We will revise the manuscript to better clarify these points and provide further context on the baselines' performance.

---

> ### Author Response · Authors · 2024-11-24
> **Response to Reviewer Z2a9 (Part 2)**
>
> > **Q3.** Why don’t you compare against methods in the "Existing Attacks against Web Agents" section?
>
> Thank you for raising this question. The methodologies in the "Existing Attacks against Web Agents" section focus on different goals, making direct comparisons less meaningful in our setting. Specifically:
>
> Yang et al. (2024) and Wang et al. (2024) explore backdoor attacks, which involve embedding malicious functionality during the training or deployment phase, whereas our work targets prompt injection attacks in a black-box setting to manipulate agent behavior dynamically.
>
> Wu et al. (2024b) and Liao et al. (2024) investigate privacy attacks, focusing on extracting sensitive information from the agent or its training data. These goals are distinct from our objective of changing the agent's behavior to perform specific targeted actions.
>
> Wu et al. (2024a) examines image modification attacks, which aim to alter the visual input to deceive vision-language models. This is fundamentally different from our focus on text-based HTML prompt injection and does not align with our attack scenario.
>
> While the existing methods provide valuable insights into related attack vectors, their objectives and methodologies diverge significantly from our focus on controllable black-box prompt injection attacks for influencing agent actions. Therefore, comparing our method against these works would not provide a fair or informative evaluation of our contributions.
>
> > **Q4.** Discussion of more ablations would clarify your approach.
>
> Thank you for the insightful suggestion. Following your advice, we have incorporated additional ablations for comparison. Specifically, we evaluated the attack success rates of (1) heuristic-based human-written adversarial strings, (2) GPT-generated adversarial strings obtained from Algorithm 2, (3) SFT, and (4) our proposed AdvWeb framework (SFT + DPO). The results show that human-written and GPT-generated strings generally achieve lower attack success rates (ASR) compared to SFT, except in the cooking domain, likely due to the simpler nature of tasks in that domain. Notably, our AdvWeb framework consistently achieves the highest ASR across all domains, demonstrating the effectiveness of combining SFT and DPO in generating adversarial strings.
>
> | Methods       | Finance | Medical | Housing | Cooking |
> |---------------|---------|---------|---------|---------|
> | Human written | 16.0    | 8.0     | 16.0    | 72.0    |
> | GPT generated | 63.6    | 67.3    | 19.6    | 96.7    |
> | SFT           | 94.8    | 74.8    | 59.2    | 49.2    |
> | AdvWeb     | 100.0   | 94.4    | 97.6    | 98.0    |
>
> These results highlight the superior performance of AdvWeb, particularly in complex domains like finance and medical, where the combination of SFT and DPO significantly boosts the attack's effectiveness.
>
> > **Q5.** How expensive is the training and optimization process?
>
> We used the mistralai/Mistral-7B-Instruct-v0.2 model for adversarial prompter training. The attack generation pipeline involves two stages:
>
> 1. SFT: This step leverages the Together fine-tuning API, taking less than $5 for training.
>
> 1. DPO: The DPO process, which refines the model based on feedbacks, takes 3 hours for 100 optimization steps on a single NVIDIA A6000 GPU.
>
> We will include this cost analysis in the revised paper to highlight the efficiency and scalability of our attack framework.
>
> > **Q6.** The title focuses on "VLM-Powered Web Agents," but the attacks target HTML parsing, not multimodality. I suggest revising the title.
>
> Thank you for pointing this out. We agree that the current title may lead to confusion regarding the scope of the attacks. To better reflect the focus of our work, we will revise the title in the updated manuscript by removing "VLM-powered."
>
> > **Q7.** The transfer results aren't as convincing as their main results - especially since the ASR is quite varied across different domains, achieving 0% transfer on probably the most compelling domain for their threat model (online finance tasks).
>
> We appreciate the reviewer's observation. The poor transferability in almost all domains, including finance, highlights the inherent limitations of simple transfer-based approaches. Adversarial strings optimized on one model often fail to generalize due to variations in model architectures, training data, and decision-making heuristics. This finding underscores the need for a dedicated black-box optimization process like ours.
>
> > **Q8.** Some claims in the abstract and introduction seem too strong. For example, VLMs “revolutionized” web agents, and HTML injection is “nearly impossible to detect.”
>
> Thank you for pointing out the need for stronger justification of our claims. We agree that the phrasing in these statements can be improved for precision and nuance. We will revise the statements to ensure that our claims are both accurate and appropriately supported.

---

> > ### Comment · Reviewer_Z2a9 · 2024-11-25
> >
> > > Q3: existing attacks
> >
> > How difficult would it have been to adapt their attack methodologies to your goals? Or yours to theirs? I don't feel like I have strong like:like comparisons between AdvWeb and other methods, which makes me underconfident that I can make claims about how successful or dangerous your method is.
> >
> > > Q4: Ablations
> >
> > These are very informative! I'm grateful for your work evaluating these ablations. Just to check, are these ablations cumulative as you go down the table, or does SFT just achieve such a high ASR on its own on financial tasks?
> >
> > > Q5: Cost, Q6: Title, and Q8: Strong Claims
> > > "We will include this cost analysis in the revised paper"
> > > "removing "VLM-powered."
> > > "We will revise the statements to ensure that our claims are both accurate and appropriately supported."
> >
> > These will be improvements to the paper. I look forward to reading the revised version
> >
> > > Q7: transfer
> > > "underscores the need for a dedicated black-box optimization process like ours."
> >
> > Perhaps, but it still makes it very hard for me to get a handle on ~any reasonable comparison I would want to make to understand how effective your method is. I look forward to the forthcoming new baselines.

---

> ### Author Response · Authors · 2024-11-24
> **Response to Reviewer Z2a9 (Part 3)**
>
> > **Q9.** What signal do you optimize against from the target model? Does the victim model refuse?
>
> We directly test the generated adversarial string on the target black-box agent (e.g., GPT-4V-powered SeeAct) and optimize the success rate of eliciting the desired adversarial behavior (e.g., purchasing NVIDIA stock instead of Microsoft stock). The optimization signal comes from binary feedback (success or failure) received after evaluating whether the adversarial string caused the agent to perform the intended action. The victim model rarely refuses since our task does not involve harmful content, unlike certain attack scenarios in baselines like GCG. Our approach focuses on subtly misleading the agent rather than prompting it to take explicitly unsafe or malicious actions, ensuring higher success rates and maintaining stealth.
>
> > **Q10.** Why does transfer from GPT-4V to Gemini fail in finance tasks?
>
> Upon manual inspection, we observe that the low transfer success rate across all domains, including finance, arises from the distinct ways GPT-4V and Gemini process and interpret adversarial injections. Each model exhibits unique patterns of successful adversarial strings.
>
> > **Q11.** How were tasks selected from Mind2Web? What criteria judged their criticality?
>
> We appreciate the reviewer’s question regarding task selection. The selection process involved the following considerations, mainly through manual inspection:
>
> 1. Severity of Consequences: While severeness can be subjective, we focused on tasks where incorrect actions could lead to substantial financial, personal, or operational harm (e.g., buying stocks, incorrect financial transactions, or healthcare-related actions). These were chosen to align with the potential high-stakes applications of web agents.
>
> 1. Representation of SeeAct: Tasks were selected to be representative of the SeeAct framework’s capabilities, demonstrating its practical applications in real-world scenarios.
>
> 1. Diversity: We selected tasks across different domains (e.g., finance, cooking, housing) to showcase the diversity of applications and the effectiveness of the proposed method in varying contexts.

---

> > ### Comment · Reviewer_Z2a9 · 2024-11-25
> >
> > > Q9: Optimisation target
> > > "The optimization signal comes from binary feedback"
> >
> > Helpful and interesting to know. Thank you for clarifying.
> >
> > > Q10: Transfer
> > > "arises from the distinct ways GPT-4V and Gemini process and interpret adversarial injections"
> > That's interesting, I'd be curious to inspect examples in an appendix if possible.
> >
> > > Q11: Selection
> > > "could lead to substantial financial, personal, or operational harm"
> >
> > These are reasonable criteria. Could this detail be added to the paper, perhaps with examples from each category?

---

> ### Comment · Reviewer_Z2a9 · 2024-11-25
>
> > Q1 In the housing domain, the potential economic loss is significant, as tasks often involve selecting properties where price and location are critical factors
>
> I'm grateful you provided this additional detail! I think it's with the addition of information like this to the paper, or at least an appendix, that would help a reader judge how important this work is.
>
> > Q2: Baselines
>
> Thanks for these extra details about why the baselines you selected perform poorly. I think the first two points in the authors' overall comment on the paper make me optimistic, but I'll wait and see for them to be included in the revised paper.

---

### Official Review · Reviewer_9jHh · 2024-11-02

**Soundness:** 2
**Presentation:** 3
**Contribution:** 3
**Rating:** 6
**Confidence:** 4

**Summary:**

This paper presents AdvWeb, a framework that injects invisible adversarial strings into websites to attack VLM-powered web agents. The technique uses Direct Policy Optimization (DPO) to optimize a prompter model that generates effective adversarial strings. The key claim is that the attack transfers effectively across different targets and settings. The main technical contribution is the technique for injecting invisible adversarial strings onto websites and using DPO to optimize a prompter model to make these strings work well.

I am excited about this paper's potential, as it addresses an important practical problem. However, several issues currently prevent me from advocating for acceptance. If these issues are adequately addressed in the rebuttal, I would revise my recommendation.

**Strengths:**

+: handling an important problem, web-use agents with black box access, quite close to what would happen in practice i.e., the threat model seems realistic, point about the stealthiness is important
+ high ASR, but how is the ASR measured?
+ The paper seems to be clear and well written
+ Good use of train and test tasks

**Weaknesses:**

High priority:

- ** One weakness is that because the HTML isn't rendered, if you had a purely computer image based agent, it would not work. It would have been better to study stealthiness in that setting. I.e, studying agents that _don't_ take in the HTML. Please at least discuss this limitation in the paper.
- ** Concerns about Baselines. I feel like the baseline should probably be someone adding a GCG string to a website HTML as a prompt injection attack, rather than as a jailbreak? You can use transfer there as you did. Did you use jailbreaks that work on GPT-4v? I think there should be working jailbreaks (e.g., Pliny's Godmode Jailbreak or Many Shot jailbreaking). I find the fact that no other attacks work pretty suspicious, and wonder how hard you tried to optimize them? FWIW, I am also sympthatetic to the view that the prompt injection is just a new setting, in which case, a human written baseline, a prompting online, SFT, and SFT + DPO baseline may be more appropriate, which is already later in the paper (but framed differently)
- ** "The results, as shown in Table 3, demonstrate that the ASR remains high when we change the injection position or HTML field, with ASR being 71.0% and 97.0%, respectively." is a misclaim, given that it depends a lot on the domain with finance performing poorly. The overall claim is OK, but please rewrite the claim to be correct. In general, you need to go through the paper and make sure you aren't overclaming.

Medium priority:
- * Please add more examples of the attacks found in the main paper, and mention diveristy. Do we have model collapse?
- * Please move Algorithm 2 into the main paper, its mentioned several times there. It's also not an algorithm, its a function. Please fix this.
- * You need to explain how you are doing the labelling in Algorithm 1, I assume you use an LLM but that seems important? Or do you just check the action taken is hard?
- * I appreciate this is hard, but I'd love to see whether the attack works on Claude Sonnet 3.5. Claude models tend to be more robust, so thats why I am curious.

Low priority:
- "innovative reinforcement learning (RL)-based attacking pipeline, tailored to solve these challenges effectively". Please drop "innovative". Similarly "advanced" attacking framework, drop advanced. Please fix the language.

**Questions:**

- * I am confused about why the different between SFT and DPO varies so much by domain in figure 3. Can you help me understand this? This makes me suspicious.
    - Do you need RL? What about a prompting based baseline?
    - I did not follow "We also fix the injection position at the ground truth HTML element e to further reduce the search space"
    - In Algorithm 2, you are using the positive examples twice, both for SFT and then also for DPO. I'm curious whether this introduces some bias, and wondering if you have the comparison without this. This should be easy to do.
    - I'd be curious for an adversarial prompter model scaling study, how it varies with size.
    - Is there any overlap between the train and test tasks? How does the model algorithm performance scale with the number of train tasks?

---

> ### Author Response · Authors · 2024-11-24
> **Response to Reviewer 9jHh (Part 1)**
>
> We thank the reviewer for the detailed and constructive feedback. We appreciate the acknowledgment of our work’s relevance and strengths, including its focus on a realistic threat model, the importance of stealthiness, and the high attack success rate (ASR). Below, we address each of the concerns and questions raised.
>
> > **Q1.** The HTML isn’t rendered, so if you had a purely computer image-based agent, it would not work. Study stealthiness in that setting or at least discuss this limitation.
>
> We appreciate the reviewer highlighting this limitation. Currently, state-of-the-art web agents, such as GPT-4V-powered SeeAct, predominantly rely on HTML input alongside visual context for decision-making. This is because HTML provides a rich, structured representation of the webpage, enabling agents to efficiently parse and act on its content. Many advanced web agents focus on leveraging HTML for tasks like form filling, navigation, and interaction, making this a practical and relevant setting for our study.
>
> However, we acknowledge that purely vision-based agents, which operate exclusively on rendered webpage images, may not be as susceptible to our current attack framework. We will discuss this limitation in the revised paper and explore extending AdvWeb to target vision-based agents in future work.
>
> > **Q2.** Concerns about baselines: I find the fact that no other attacks work pretty suspicious. How hard did you try to optimize them? Did you test jailbreaks that work on GPT-4V?
>
> We took significant effort to adapt existing baselines to our setting:
>
> 1. GCG and AutoDAN: Optimized adversarial strings against a strong open-source VLM (LLaVA-NeXT) and transferred them to GPT-4V.
>
> 1. COLD-Attack: Adapted constrained decoding for stealthy adversarial prompts, again transferring optimized prompts from LLaVA-NeXT.
>
> 1. Catastrophic Jailbreak: Varied decoding parameters to manipulate outputs within the black-box setting.
>
> Despite significant effort to adapt these methods, the poor performance of these baselines is likely due to two key factors:
>
> 1. Different Optimization Objectives: Most prior attacks aim to elicit affirmative responses or bypass filters, whereas our attack requires preserving the original response structure while altering specific target elements or actions. This dual requirement makes the optimization problem more complex.
>
> 1. Limited Transferability: Adversarial strings optimized against white-box models like LLaVA-NeXT struggle to transfer effectively to proprietary black-box models like GPT-4V, particularly in targeted attack scenarios where the target action or element must be precisely controlled.
>
> We will include additional explanations and new baselines (e.g., human-written prompts, direct GPT prompting, and SFT) to clarify these differences.
>
>
> > **Q3.** A human-written baseline, direct GPT prompting, SFT, and SFT+DPO baseline may be more appropriate.
>
> Thank you for the insightful suggestion. Following your advice, we have incorporated additional baselines for comparison. Specifically, we evaluated the attack success rates of (1) heuristic-based human-written adversarial strings, (2) GPT-generated adversarial strings obtained from Algorithm 2, (3) SFT, and (4) our proposed AdvWeb framework (SFT + DPO). The results show that human-written and GPT-generated strings generally achieve lower attack success rates (ASR) compared to SFT, except in the cooking domain, likely due to the simpler nature of tasks in that domain. Notably, our AdvWeb framework consistently achieves the highest ASR across all domains, demonstrating the effectiveness of combining SFT and DPO in generating adversarial strings.
>
> | Methods       | Finance | Medical | Housing | Cooking |
> |---------------|---------|---------|---------|---------|
> | Human written | 16.0    | 8.0     | 16.0    | 72.0    |
> | GPT generated | 63.6    | 67.3    | 19.6    | 96.7    |
> | SFT           | 94.8    | 74.8    | 59.2    | 49.2    |
> | AdvWeb     | 100.0   | 94.4    | 97.6    | 98.0    |
>
> These results highlight the superior performance of AdvWeb, particularly in complex domains like finance and medical, where the combination of SFT and DPO significantly boosts the attack's effectiveness.
>
> > **Q4.** Rewrite the claim on ASR to reflect domain-specific performance.
>
> Thank you for pointing this out. We agree that the performance varies across domains, and the claim should be more nuanced. In our revision, we will explicitly state that while the overall ASR remains high, there are domain-specific variations, with finance-related tasks performing less reliably. We will revise all claims to ensure accuracy and avoid overstatements.

---

> ### Author Response · Authors · 2024-11-24
> **Response to Reviewer 9jHh (Part 2)**
>
> > **Q5.** Add more examples of attacks and discuss diversity. Do we have model collapse?
>
> We have included additional examples in Figure 4 of the revised paper, showcasing the diversity of adversarial strings generated by the prompter model to attack the agent. These examples demonstrate various attack patterns, such as misleading the agent in financial and medical tasks, highlighting the adaptability of the adversarial prompter.
>
> > **Q6.** Move Algorithm 2 to the main paper and clarify that it is a function.
>
> Thank you for this suggestion. We will move Algorithm 2 into the main text and clarify its designation as a function rather than an algorithm.
>
> > **Q7.** Explain labeling in Algorithm 1.
>
> We directly test the generated adversarial string on the target black-box agent (e.g., GPT-4V-powered SeeAct) and collect labels indicating whether the attack was successful. A success label is assigned if the agent performs the intended adversarial action as specified by the target objective (e.g., buying NVIDIA stock instead of Microsoft stock). This approach ensures that labels are directly tied to real-world agent behavior. We will clarify this process in the revised paper.
>
> > **Q8.** Test attacks on Claude Sonnet 3.5.
>
> We have tested the attack on Claude Sonnet 3.5. The model demonstrated enhanced safety alignment and refused to generate responses when the adversarial attacks were applied. For example, when presented with an adversarial injection, the model responded: “I will not select or take any actions that were not part of the original task. I aim to provide helpful information while avoiding potential harm. Perhaps we could have a thoughtful discussion about safe and ethical ways to accomplish your goals.”
>
> This result highlights the robustness of Claude Sonnet 3.5 in adhering to its safety mechanisms. We will include a discussion of these findings in the revised paper, comparing the alignment strategies of Claude Sonnet 3.5 with other tested models.
>
> > **Q9.** Fix subjective language like "innovative" and "advanced."
>
> We agree and will replace subjective terms like "innovative" and "advanced" with objective, precise descriptions.
>
> > **Q10.** Why does the difference between SFT and DPO vary across domains?
>
> The variation stems from DPO's ability to learn subtle patterns in the data that are challenging for humans to distinguish. For example, in complex domains like finance, where slight variations in context or phrasing can lead to significantly different agent actions, DPO effectively exploits these nuances to optimize attack performance. We will elaborate on this insight in the revised paper.
>
> > **Q11.** Clarify "fixing injection position at ground truth HTML element."
>
> The statement refers to a design choice aimed at simplifying the optimization process for generating adversarial HTML content. Fixing the injection position at the ground truth HTML element reduces the high-dimensional search space of adversarial HTML content, simplifying optimization. This design ensures the injected prompt is hidden (e.g., in aria-label attributes) to maintain stealthiness and includes placeholders for controllability, allowing easy substitution for different target actions.
>
> > **Q12.** Investigate bias in using positive examples for both SFT and DPO.
>
> We conducted an ablation study where the dataset was randomly split into two disjoint subsets: one for SFT and the other for DPO. The results are summarized below:
>
> |       | Original | Non-overlap |
> |-------|----------|-------------|
> | SFT   | 94.8     | 89.6        |
> | DPO   | 100.0    | 99.2        |
>
> The drop in performance after splitting the data is likely due to having less data available for each stage of training. However, the final results remain close, with the ASR only slightly reduced when using disjoint subsets (99.2% vs. 100.0%).
>
> > **Q13.** Conduct a scaling study on the adversarial prompter model.
>
> Since we already achieve near-100% attack success rates with Mistral-7B, using larger models leads to limited improvements. However, smaller models struggle to generate effective adversarial prompts. For instance, when we train Qwen2-1.5B-Instruct, the attack success rate drops to 0%, highlighting that small models are insufficient for this task. This demonstrates that a mid-sized model strikes the right balance between computational efficiency and effectiveness. We will include these results and insights in the revised paper.
>
> > **Q14.** Is there any overlap between train and test tasks?
>
> There is no overlap between train and test tasks. We ensure strict separation to evaluate the model’s generalization.

---

> > ### Comment · Reviewer_9jHh · 2024-12-02
> >
> > Thanks, I raised my score.

---

### Official Review · Reviewer_NU19 · 2024-11-03

**Soundness:** 3
**Presentation:** 3
**Contribution:** 2
**Rating:** 5
**Confidence:** 2

**Summary:**

The paper studies the robustness of Web Agents and proposes a black-box adversarial attack AdvWeb which is based on Direct Policy Optimization (DPO). The attack efficiency is evaluated against existing SOTA web agent for several domains.

**Strengths:**

Originality: the papers proposes the first black-box attack against VLM-based web agents which is based on existing DPO Method.

Quality: the authors perform in depth analysis of their proposed attack. The limitations of the method are discussed (Appendix C).

Clarity: the paper is structured reasonably.

Significance: black-box attacks on web agents may cause significant harm, so raising awareness of such attacks is important for the LLM/VLM field.

**Weaknesses:**

1) Could you elaborate in the paper on how exactly the baselines (line 353) were adapted to your problem? Since they all achieve 0.0 on all tasks (Tables 1-2), it looks like the comparison is either not suitable or not fair. Do you have an explanation for such poor performance of the baselines?

2) See the ethics review discussion.

**Questions:**

1) Do you have any suggestions on how one can defend web-agents against attacks similar to the one that you propose?

**Details Of Ethics Concerns:**

I would like to recommend the authors to avoid using existing company (stocks) names where it is not really required (Figure 1 and further in the text). I think, in the examples the names can be replaced with e. g. “Company A” and “Company B” or "Stocks A" and "Stocks B" without any loss for the paper motivation and content. I think, the reasoning for using real names might have been to show an actual example of the problem tackled in the paper. However, I think that using real names (especially in the context of an adversarial attack) does not contribute anything to the discussion and may potentially lead to uninvited implications.

---

> ### Author Response · Authors · 2024-11-24
> **Response to Reviewer NU19**
>
> We appreciate the reviewer's thoughtful feedback and recognition of the originality and significance of our work. Below, we address the specific concerns raised:
>
> > **Q1.** How exactly were the baselines (line 353) adapted to your problem? Do you have an explanation for such poor performance of the baselines?
>
> Thank you for the question. Since there is no existing black-box attack against web agents in our setting, we adapted the following attacks to our targeted black-box attack scenario:
>
> 1. GCG: This white-box attack optimizes adversarial suffix strings using token-level gradients from the target model. In our setting, where gradient access to the black-box agent is unavailable, we optimized adversarial strings against a strong open-source VLM, LLaVA-NeXT, and transferred them to the GPT-4V-based web agent.
>
> 1. AutoDAN: AutoDAN: This white-box attack uses genetic algorithms to optimize adversarial suffixes by leveraging logits from the target model. We optimized adversarial prompts against LLaVA-NeXT and transferred them to the black-box agent.
>
> 1. COLD-Attack: This method generates stealthy adversarial prompts by leveraging energy-based constrained decoding, requiring access to model gradients. Similarly, we optimized adversarial strings on LLaVA-NeXT and transferred them.
>
> 1. Catastrophic Jailbreak: This black-box attack manipulates decoding variations to disrupt alignment, focusing on system prompts and decoding parameters. Since our setting does not allow access to system prompts, we adapted the decoding parameter variation.
>
> The poor performance of these baselines can be attributed to two key factors. First, the optimization goal in our attack is different from prior works, which typically aim to elicit affirmative responses or bypass filters. In contrast, our attack requires preserving the original response while altering the target element or action within it, making it inherently more complex. Second, adversarial strings optimized against white-box models like LLaVA-NeXT are challenging to transfer effectively to proprietary black-box models like GPT-4V, especially in targeted attack settings.
>
> We will clarify these adaptations and the unique challenges of our setting in the revised paper.
>
> > **Q2.** Do you have any suggestions on how one can defend web agents against attacks similar to the one proposed?
>
> Thank you for raising this important question. While designing robust defenses against adversarial attacks is a non-trivial challenge, we propose the following suggestions to enhance the security of web agents:
>
> 1. Adversarial Robustness through Training: Incorporating adversarial training methods, where web agents are trained on adversarially perturbed examples, can help improve their robustness against such attacks.
>
> 1. Input Sanitization and Validation: Web agents should implement strict sanitization processes to ignore hidden HTML attributes (e.g., aria-label, id) and rely solely on visible and user-intended elements for decision-making.
>
> 1. Action Validation with Contextual Awareness: Introducing a secondary verification mechanism for critical actions, such as cross-checking proposed actions against the user’s task context, can help prevent unintended or harmful behaviors.
>
> 1. Behavioral Monitoring and Logging: Developing monitoring tools to track and analyze web agent behaviors in real time can help detect anomalies and suspicious actions, enabling timely intervention.
>
> We will include a dedicated discussion on these potential defenses in the revised paper to provide further insights into safeguarding web agents.
>
>
> > **Q3.** Avoid using existing company (stock) names where not required.
>
> Thank you for your suggestion. We will follow your suggestion and replace real company names with generic placeholders such as “Company A” and “Company B” or "Stocks A" and "Stocks B" throughout the paper, including in all figures, tables, and text.

---

> > ### Comment · Reviewer_NU19 · 2024-11-27
> > **Thank you**
> >
> > Dear authors,
> >
> > thank you for your rebuttal and for addressing the points that I have raised. After familiarizing myself with the other reviews and responses I have decided to keep my score.

---

### Author Response · Authors · 2024-11-24

We thank the reviewers for their time, thoughtful feedback, and recognition of the originality and significance of our work. We are glad that the reviewers appreciated our focus on exposing vulnerabilities in web agents and the effectiveness of our proposed AdvWeb framework. Below, we summarize the key improvements and clarifications we have made based on the reviewers' comments:

1. Baselines: We have expanded our experimental setup to include additional baselines, including human-written prompts, direct GPT prompting, and SFT, alongside SFT+DPO (AdvWeb). These comparisons further demonstrate the superior attack success rate (ASR) of our method and its adaptability across domains.

1. Experimental Details: We have clarified the implementation details of baselines, such as GCG, AutoDAN, and COLD-Attack, explaining how they were adapted to the black-box setting. We also included detailed descriptions of how adversarial strings are optimized and evaluated, addressing concerns about baseline performance and transferability challenges.

1. Threat Model and Limitations: We have expanded our discussion on the threat model, detailing practical scenarios where malicious injections could occur (e.g., visiting malicious websites, compromised trusted websites, and open-source package vulnerabilities). We also acknowledge limitations in addressing purely vision-based agents and scenarios with robust defenses like secondary authentication.

1. Defenses and Mitigation: In response to reviewers’ requests, we included a discussion on potential defenses, such as adversarial training, input sanitization, contextual action validation, and real-time behavioral monitoring, to safeguard web agents against attacks like AdvWeb.

1. Writing and Presentation: We have addressed comments on the clarity and quality of writing, fixing typos, improving figure readability, and rephrasing overly strong claims to ensure precision and accuracy.

1. New Insights and Model: Based on reviewer feedback, we tested the robustness of Claude Sonnet 3.5 and observed its enhanced alignment, which prevented adversarial behavior. This finding has been incorporated into the revised paper as a benchmark comparison.

1. Impact of Attack Scenarios: We have added detailed examples of attack scenarios, their consequences, and their implications across various domains, such as finance, medical, and housing, to enhance the understanding of the risks associated with adversarial attacks.

In summary, we have taken significant steps to address the reviewers' questions by expanding our evaluations, improving clarity, enhancing the structure of our paper, and adding more discussions. We also integrated these improvements in our revisions and we believe they will further strengthen the contribution and impact of our work.

---

### Meta-Review · Area_Chair_Lreo · 2024-12-12

**Metareview:**

This paper proposes an attacking framework (AdvWeb) against VLM-powered web agents, which mainly focuses on optimizing a prompter model for generating adversarial injections. Its findings help raise public awareness of the potential risks of deploying VLMs in real-world scenarios. However, its weaknesses include the singleness of the web agents evaluated, unfair comparison with baselines, and lack of evaluation of advanced defenses. Overall, this work does not reach an acceptable state given the raised issues.

**Additional Comments On Reviewer Discussion:**

The reviewers raise several crucial weaknesses of this work, and the authors fail to address all of them. Most reviewers point out the limitations of the evaluated baselines, both in their design goals and evaluation performance, making it difficult to provide a reasonable comparison with AdvWeb. The authors provide ablation experiments on human written, GPT generated, and SFT, but they do not further respond to Reviewer Z2a9's follow-up question about SFT. In addition, Reviewers NU19, d1HB, and 7hWf think that this work lacks consideration of defenses. Though the authors discuss potential defenses during the rebuttal, there is still no experimental evaluation provided for any defenses. These unsolved weaknesses indicate that this work needs further improvement to reach an acceptable level.

---

### Decision · Program_Chairs · 2025-01-22

Reject